# Eosin Y-catalyzed visible-light-mediated aerobic transformation of pyrazolidine-3-one derivatives

**Nejc Petek, Uroš Grošelj, Jurij Svete, Franc Požgan, Drago Kočar, Bogdan Štefane\***

1  *Faculty of Chemistry and Chemical Technology, University of Ljubljana, Večna pot 113, SI-1000 Ljubljana Slovenia*; Bogdan.stefane@fkkt.uni-lj.si

\*  Correspondence: Bogdan.stefane@fkkt.uni-lj.si; Tel.: +386-1-4798560

## 1.  Cyclic voltammetry

Cyclic voltammograms were recorded on ElectraSyn 2.0 (IKA, Staufen im Breisgau, Germany) on glassy carbon working electrode with Pt plated counter electrode and Ag wire quasi-reference electrode. Solution of **1c** (10 mM) was prepared in acetonitrile with $Bu_4NBF_4$ (50 mM) as an electrolyte. Ferrocene was added as an internal standard.

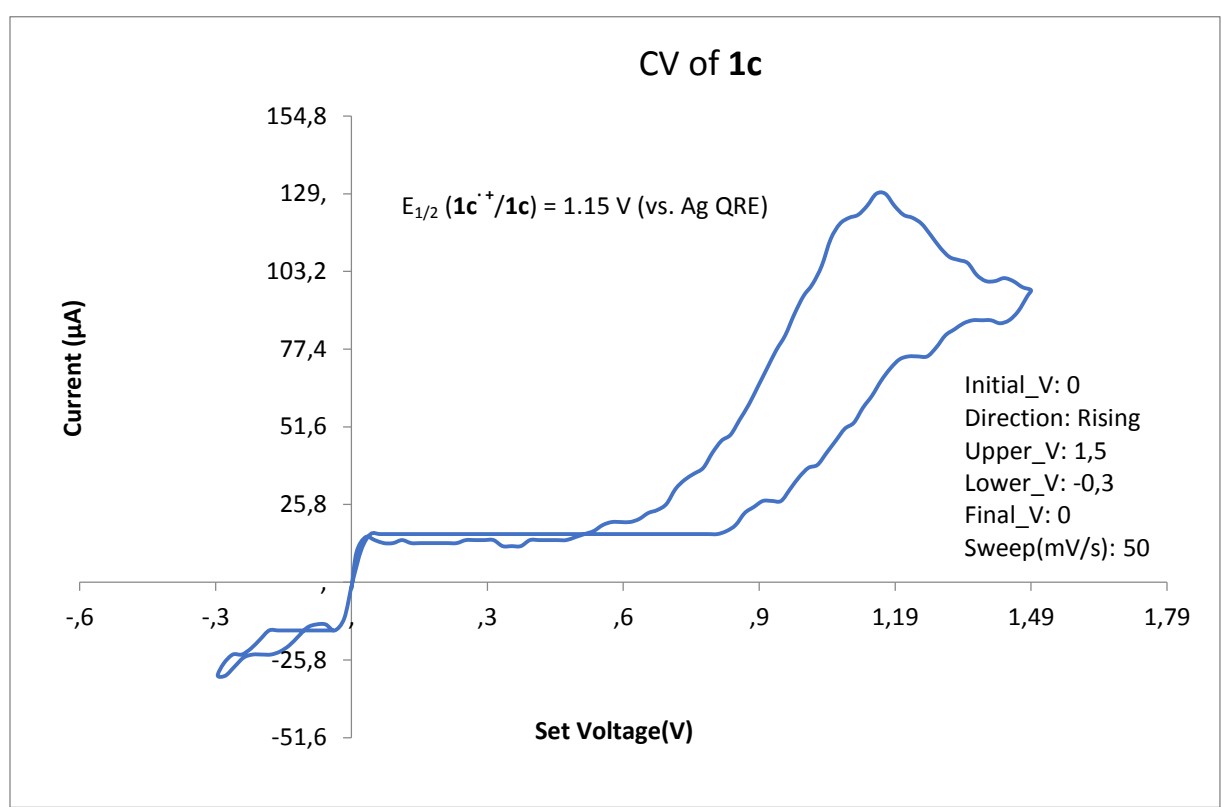

CV of **1c**

$E_{1/2}$ (**1c**$^{\cdot+}$/**1c**) = 1.15 V (vs. Ag QRE)

Initial_V: 0
Direction: Rising
Upper_V: 1,5
Lower_V: -0,3
Final_V: 0
Sweep(mV/s): 50

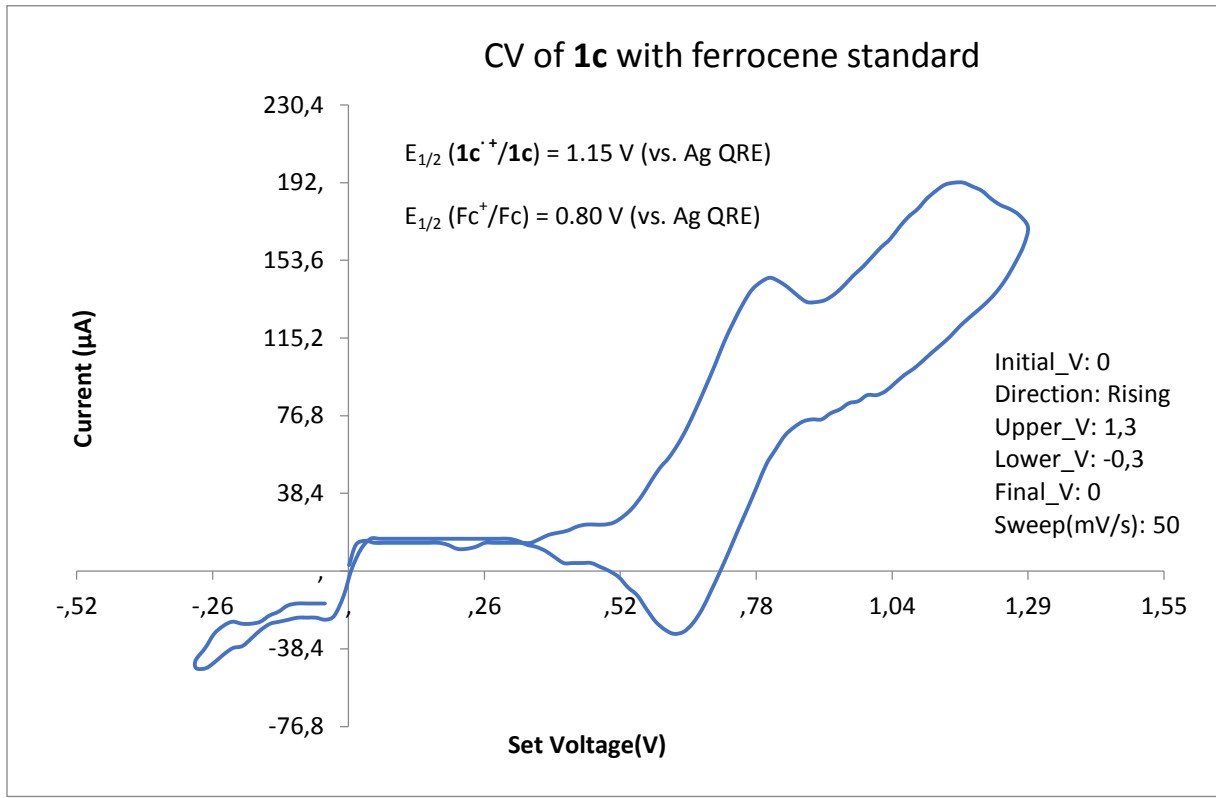

Literature potential for $Fc^+/Fc$ vs. SCE is 0.38 V in acetonitrile.

$E_{1/2}$ (**1c**$^{·+}$/**1c**) = 0.35 V (vs. Fc)

$E_{1/2}$ (**1c**$^{·+}$/**1c**) = 0.73 V (vs. SCE)

Aranzaes, J. R., Daniel, M.-C., Astruc, D. Metallocenes as references for the determination of redox potentials by cyclic voltammetry. – Permethylated iron and cobalt sandwich complexes, inhibition by polyamine dendrimers, and the role of hydroxy-containing ferrocenes. *Can. J. Chem.* **2006**, *84*, 288–299.

## 2. Absorption spectroscopy

Absorption spectra of Eosin Y-disodium salt was recorded on Cary 50 Bio UV-VIS Spectrophotometer (Agilent Technologies, Santa Clara, CA, USA). Normalized spectra recorded in acetonitrile and acetonitrile/TFA (0.5M solution as used herein for oxidation of compounds **1**) is shown.

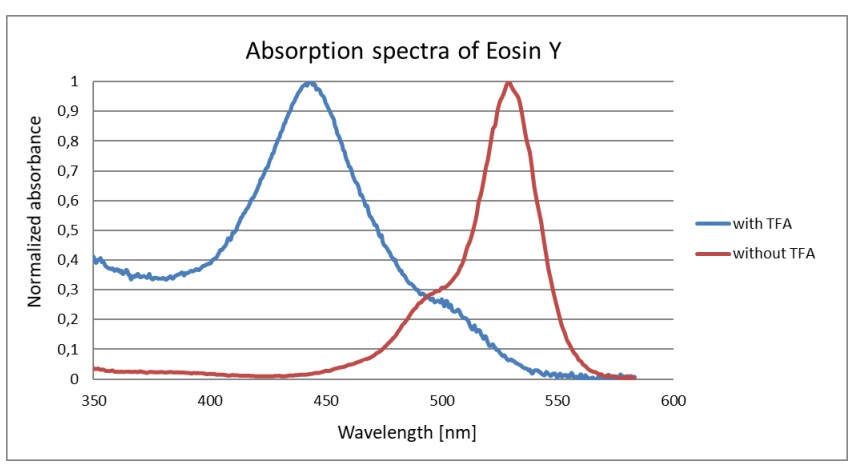

### 3. Mass spectrometry

TEMPO adduct of **1c** was detected on Agilent 6224 Accurate Mass TOF LC/MS spectrometer.

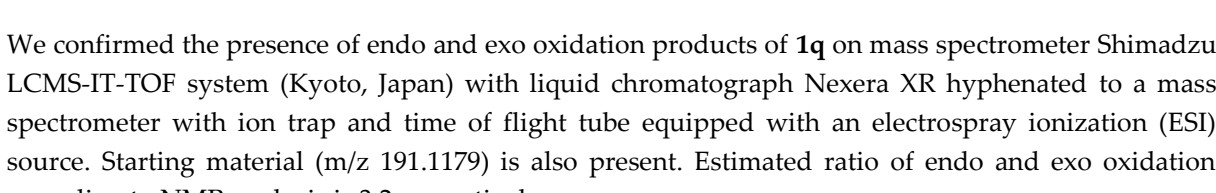

We confirmed the presence of endo and exo oxidation products of **1q** on mass spectrometer Shimadzu LCMS-IT-TOF system (Kyoto, Japan) with liquid chromatograph Nexera XR hyphenated to a mass spectrometer with ion trap and time of flight tube equipped with an electrospray ionization (ESI) source. Starting material (m/z 191.1179) is also present. Estimated ratio of endo and exo oxidation according to NMR analysis is 3:2 respectively.

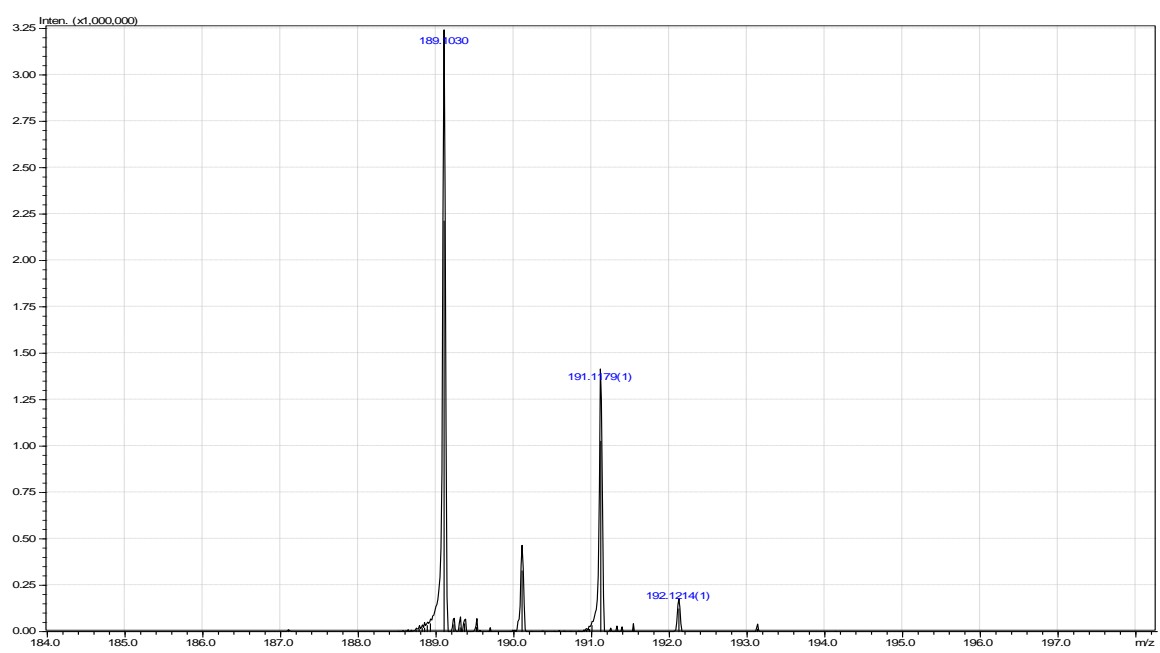

| No. | Mass | Diff (mDa) | Formula | DBE |
|-----|----------|------------|---------------|------|
| 1 | 188.0950 | 0.8 | C11 H12 N2 O | 7.0 |
| 2 | 188.0909 | 4.8 | C6 H12 N4 O3 | 3.0 |
| 3 | 188.0797 | 16.0 | C7 H12 N2 O4 | 3.0 |
| 4 | 188.1121 | 16.3 | C3 H16 N4 O5 | -2.0 |

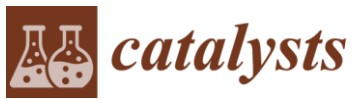 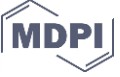

4. Characterization of compounds **1–5**.

**General Remarks**

Reactions were carried out in a commercially available SynLED Parallel Photoreactor (465–470 nm LEDs, 130–140 lm, Sigma-Aldrich, St. Louis, MO, USA), which operates at approximately 25–30 °C. The NMR spectra were recorded in deuterated solvents with Me₄Si as the internal standard on a Bruker Avance DPX 300 (Bruker, Billerica, MA, USA) and Bruker Avance III UltraShield 500 (Bruker, Billerica, MA, USA) plus instruments at 300 and 500 MHz for ¹H and at 75.5 and 126 MHz for ¹³C nuclei, respectively. Data for ¹H NMR are reported as chemical shifts (δ) in ppm, integration, multiplicity (s = singlet, d = doublet, t = triplet, q = quartet, m = multiplet), coupling constant and assignment. Data for ¹³C are reported as chemical shift (δ) in ppm. Mass spectra were recorded on Agilent 6224 Accurate Mass TOF LC/MS spectrometer (Agilent Technologies, Santa Clara, CA, USA) and IR spectra on a Bruker FTIR Alpha Platinum spectrophotometer (Bruker, Billerica, MA, USA). Melting points were determined on a Kofler hot-stage microscope and on a MPA100 OptiMelt automated melting point system (Stanford Research Systems, Sunnyvale, CA, USA). Thin-layer chromatography (TLC) was performed on aluminum backed silica plates (0.2 mm, 60 F254, Sigma-Aldrich, St. Louis, MO, USA). Visualization of TLC (254 nm, Camag, Muttenz, Switzerland) was performed by fluorescence quenching or with potassium permanganate stains. Flash column chromatography (FC) and column chromatography (CC) were performed on silica gel (particle size: 35–70 μm, Sigma-Aldrich, St. Louis, MO, USA). Commercially available compounds were used without further purification.

**Pyrazolidin-3-ones 1.**

Compounds **1a-p** were prepared according to established literature procedures [1-4].

1-Benzyl-5,5-dimethyl-pyrazolidinone (**1a**). The ¹H-NMR data is in agreement with literature. [1]

5,5-Dimethyl-1-(4-methylbenzyl)pyrazolidin-3-one (**1b**). White solid (84%); mp 121–122 °C; $\nu_{max}$/cm⁻¹ (ATR) 3159, 3047, 2971, 2923, 1690, 1514, 1444, 1420, 1385, 1340, 1293, 1247, 1204, 1169, 1107, 1080, 1057, 1024, 985, 945, 922, 888, 850, 811, 788, 746, 709, 684; $\delta_H$ (500 MHz; CDCl₃; Me₄Si) 1.35 (s, 6H), 2.34 (s, 3H), 2.40 (s, 2H), 3.73 (s, 2H), 6.61 (s, 1H), 7.13 (d, $J$ = 7.9 Hz, 2H), 7.20 (d, $J$ = 7.9 Hz, 2H); $\delta_C$ (126 MHz; CDCl₃; Me₄Si) 21.1, 25.3, 43.0, 56.6, 62.3, 129.1, 129.4, 133.9, 137.5, 174.0; HRMS (ESI): MH⁺, found 219.1492. [C₁₃H₁₉N₂O]⁺ requires 219.1492.

1-(4-Chlorobenzyl)-5,5-dimethylpyrazolidin-3-one (**1c**). White solid (99%); mp 155–156 °C; $\nu_{max}$/cm⁻¹ (ATR) 3143, 3049, 2974, 1690, 1596, 1488, 1450, 1411, 1385, 1373, 1337, 1289, 1245, 1206, 1167, 1113, 1089, 1051, 1016, 989, 972, 846, 922, 889, 851, 807, 787, 749, 707, 687, 664; $\delta_H$ (500 MHz; CDCl₃; Me₄Si) 1.35 (s, 6H), 2.39 (s, 2H), 3.76 (s, 2H), 6.82 (s, 1H), 7.26 (d, $J$ = 8.5 Hz, 2H), 7.31 (d, $J$ = 8.4 Hz, 2H); $\delta_C$ (126 MHz; CDCl₃; Me₄Si) 25.4, 42.7, 56.4, 62.6, 128.8, 130.5, 133.5, 135.6, 174.3; HRMS (ESI): MH⁺, found 239.0946. [C₁₂H₁₆ClN₂O]⁺ requires 239.0946.

4-((5,5-Dimethyl-3-oxopyrazolidin-1-yl)methyl)benzonitrile (**1d**). White solid (90%); mp 156–158 °C; $\nu_{max}$/cm⁻¹ (ATR) 3153, 3054, 2974, 2858, 2228, 1690, 1610, 1505, 1468, 1448, 1418, 1374, 1339, 1293, 1248, 1209, 1167, 1111, 1079, 1049, 1023, 990, 973, 949, 925, 893, 860, 823, 793, 753, 689; $\delta_H$ (500 MHz; CDCl₃; Me₄Si) 1.35 (s, 6H), 2.39 (s, 2H), 3.86 (s, 2H), 7.25 (s, 1H), 7.47 (d, $J$ = 8.2 Hz, 2H), 7.63 (d, $J$ = 8.2 Hz, 2H); $\delta_C$ (126 MHz; CDCl₃; Me₄Si) 25.4, 42.6, 57.0, 62.9, 111.6, 118.7, 129.7, 132.4, 142.9, 174.7; HRMS (ESI): MH⁺, found 230.1290. [C₁₃H₁₆N₃O]⁺ requires 230.1288.

1-(4-Methoxy-benzyl)-5,5-dimethyl-pyrazolidin-3-one (**1e**). The ¹H-NMR data is in agreement with literature. [2]

5,5-Dimethyl-1-(4-nitrobenzyl)pyrazolidin-3-one (**1f**). Pale yellow solid (85%); mp 186–187 °C; $\nu_{max}$/cm$^{-1}$ (ATR) 3141, 3061, 2977, 2852, 1693, 1602, 1513, 1450, 1374, 1350, 1333, 1292, 1250, 1208, 1169, 1112, 1074, 1044, 1016, 985, 953, 926, 895, 873, 857, 832, 818, 755, 738, 697, 650; $\delta_H$ (500 MHz; CDCl$_3$; Me$_4$Si) 1.36 (s, 6H), 2.40 (s, 2H), 3.91 (s, 2H), 7.25 (s, 1H), 7.53 (d, $J$ = 8.8 Hz, 2H), 8.20 (d, $J$ = 8.8 Hz, 2H); $\delta_C$ (126 MHz; CDCl$_3$; Me$_4$Si) 25.5, 42.5, 56.7, 63.0, 123.8, 129.7, 144.9, 147.5, 174.7; HRMS (ESI): MH$^+$, found 250.1187. [C$_{12}$H$_{16}$N$_3$O$_3$]$^+$ requires 250.1186.

1-(2,6-Dichlorobenzyl)-5,5-dimethylpyrazolidin-3-one (**1g**). White solid (93%); mp 154–155 °C; $\nu_{max}$/cm$^{-1}$ (ATR) 3143, 3052, 2969, 1697, 1581, 1561, 1465, 1432, 1381, 1369, 1335, 1288, 1242, 1196, 1165, 1109, 1090, 1054, 1006, 967, 940, 922, 891, 843, 774, 762, 682, 625; $\delta_H$ (500 MHz; CDCl$_3$; Me$_4$Si) 1.41 (s, 6H), 2.43 (s, 2H), 4.15 (s, 2H), 6.83 (s, 1H), 7.18 (t, $J$ = 8.0 Hz, 1H), 7.32 (d, $J$ = 8.0 Hz, 2H); $\delta_C$ (126 MHz; CDCl$_3$; Me$_4$Si) 25.2, 42.6, 51.5, 63.4, 128.8, 129.7, 132.4, 137.0, 174.7; HRMS (ESI): MH$^+$, found 273.0556. [C$_{12}$H$_{15}$Cl$_2$N$_2$O]$^+$ requires 273.0556.

5,5-Dimethyl-1-(3,4,5-trimethoxybenzyl)pyrazolidin-3-one (**1h**). White solid (98%); mp 132–133 °C; $\nu_{max}$/cm$^{-1}$ (ATR) 3445, 3332, 3154, 3071, 2966, 2931, 2839, 1673, 1590, 1506, 1449, 1417, 1377, 1328, 1229, 1178, 1121, 1048, 1007, 962, 925, 886, 832, 796, 781, 748, 669, 613; $\delta_H$ (500 MHz; CDCl$_3$; Me$_4$Si) 1.37 (s, 6H), 2.42 (s, 2H), 3.72 (s, 2H), 3.83 (s, 3H), 3.87 (s, 6H), 6.55 (s, 2H), 6.73 (s, 1H); $\delta_C$ (126 MHz; CDCl$_3$; Me$_4$Si) 25.2, 43.1, 56.1, 57.2, 60.9, 62.3, 105.8, 132.6, 137.4, 153.4, 174.0; HRMS (ESI): MH$^+$, found 295.1652. [C$_{15}$H$_{23}$N$_2$O$_4$]$^+$ requires 295.1652.

1-(2-Naphthylmethyl)-5,5-dimethyl-pyrazolidinone (**1i**). The $^1$H-NMR data is in agreement with literature. [1]

Benzyl (1-benzyl-5,5-dimethyl-3-oxopyrazolidin-4-yl)carbamate (**1j**). The $^1$H-NMR data is in agreement with literature. [5]

5,5-Dimethyl-1-pyridin-3-ylmethyl-pyrazolidin-3-one (**1k**). The $^1$H-NMR data is in agreement with literature. [4]

1-(Furan-2-ylmethyl)-5,5-dimethylpyrazolidin-3-one (**1l**). Colorless oil (84%); $\nu_{max}$/cm$^{-1}$ (ATR) 3189, 2973, 1682, 1505, 1421, 1387, 1371, 1330, 1245, 1223, 1147, 1115, 1077, 1014, 974, 922, 885, 800, 734, 663; $\delta_H$ (500 MHz; CDCl$_3$; Me$_4$Si) 1.33 (s, 6H), 2.30 (s, 2H), 3.82 (s, 2H), 6.28 (d, $J$ = 3.1 Hz, 1H), 6.33 (dd, $J$ = 3.2, 1.9 Hz, 1H), 7.39 (d, $J$ = 1.7 Hz, 1H), 7.84 (s, 1H); $\delta_C$ (126 MHz; CDCl$_3$; Me$_4$Si) 25.2, 42.7, 49.6, 62.5, 109.5, 110.5, 142.6, 150.4, 174.6; HRMS (ESI): MH$^+$, found 195.1125. [C$_{10}$H$_{15}$N$_2$O$_2$]$^+$ requires 195.1128.

1-Ethyl-5,5-dimethylpyrazolidin-3-one (**1m**). The $^1$H-NMR data is in agreement with literature. [1]

Benzyl [(4$RS$,5$RS$)-1-benzyl-3-oxo-5-(propan-2-yl)pyrazolidin-4-yl]carbamate (**1n**). The $^1$H-NMR data is in agreement with literature. [5]

5,5-Dimethyl-1-(3-phenylpropyl)pyrazolidin-3-one (**1o**). White solid (77%); mp 130–132 °C; $\nu_{max}$/cm$^{-1}$ (ATR) 3051, 2964, 2848, 1687, 1649, 1491, 1453, 1415, 1385, 1340, 1239, 1124, 1024, 944, 922, 887, 796, 749, 725, 702, 679; $\delta_H$ (500 MHz; CDCl$_3$; Me$_4$Si) 1.22 (s, 6H), 1.83 (p, $J$ = 7.3 Hz, 2H), 2.32 (s, 2H), 2.62 (t, $J$ = 7.2 Hz, 2H), 2.69 (t, $J$ = 7.5 Hz, 2H), 7.14 – 7.23 (m, 3H), 7.24 – 7.33 (m, 2H), 8.80 (s, 1H); $\delta_C$ (126 MHz; CDCl$_3$; Me$_4$Si) 25.2, 29.3, 33.0, 42.9, 51.7, 62.5, 125.8, 128.3, 128.5, 141.7, 175.1; HRMS (ESI): MH$^+$, found 233.1651. [C$_{14}$H$_{21}$N$_2$O]$^+$ requires 233.1648.

5,5-Dimethyl-1-propylpyrazolidin-3-one (**1p**). Colorless oil (50%); $\nu_{max}$/cm$^{-1}$ (ATR) 3182, 2966, 1681, 1466, 1366, 1244, 1026, 888, 784; $\delta_H$ (500 MHz; CDCl$_3$; Me$_4$Si) 0.94 (t, $J$ = 7.4 Hz, 3H), 1.26 (s, 6H), 1.46 –

1.58 (m, 2H), 2.34 (s, 2H), 2.54 – 2.63 (m, 2H), 8.61 (s, 1H); $\delta_C$ (126 MHz; CDCl$_3$; Me$_4$Si) 11.6, 21.4, 25.1, 42.9, 54.5, 62.4, 175.0; HRMS (ESI): MH$^+$, found 157.1335. [C$_8$H$_{17}$N$_2$O]$^+$ requires 157.1335.

1-Ethyl-5-phenylpyrazolidin-3-one (**1q**). White solid (98%); mp 99–101 °C; $\nu_{max}$/cm$^{-1}$ (ATR) 2982, 2843, 1679, 1496, 1455, 1427, 1373, 1334, 1163, 1116, 1098, 1077, 1026, 986, 908, 872, 818, 800, 757, 702, 652, 636; $\delta_H$ (500 MHz; CDCl$_3$; Me$_4$Si) 1.13 (t, $J$ = 7.2 Hz, 3H), 2.55 (dd, $J$ = 16.8, 8.9 Hz, 1H), 2.66 (dq, $J$ = 12.0, 7.0 Hz, 1H), 2.85 (dq, $J$ = 12.0, 7.3 Hz, 1H), 3.01 (dd, $J$ = 16.8, 8.5 Hz, 1H), 4.10 (t, $J$ = 8.7 Hz, 1H), 7.28 – 7.43 (m, 5H), 8.49 (s, 1H); $\delta_C$ (126 MHz; CDCl$_3$; Me$_4$Si) 12.9, 39.4, 52.8, 67.5, 127.0, 127.9, 128.7, 140.1, 173.0; HRMS (ESI): MH$^+$, found 191.1180. [C$_{11}$H$_{15}$N$_2$O]$^+$ requires 191.1179.

1-Ethyl-5-methylpyrazolidin-3-one (**1r**) was prepared according to a literature procedure (omitting the hydrobromide formation). [6]

**Azomethine imines 2.**

(*Z*)-1-Benzylidene-5,5-dimethyl-3-oxopyrazolidin-1-ium-2-ide (**2a**). Prepared from **1a** (61%). The $^1$H-NMR data is in agreement with literature. [7]

(*Z*)-5,5-Dimethyl-1-(4-methylbenzylidene)-3-oxopyrazolidin-1-ium-2-ide (**2b**). Prepared from **1b** (63%). The $^1$H-NMR data is in agreement with literature. [8]

(*Z*)-1-(4-Chlorobenzylidene)-5,5-dimethyl-3-oxopyrazolidin-1-ium-2-ide (**2c**). Prepared from **1c** (62%). The $^1$H-NMR data is in agreement with literature. [8]

(*Z*)-1-(4-Cyanobenzylidene)-5,5-dimethyl-3-oxopyrazolidin-1-ium-2-ide (**2d**). Prepared from **1d** (48%). The $^1$H-NMR data is in agreement with literature. [8]

(*Z*)-1-(4-Methoxybenzylidene)-5,5-dimethyl-3-oxopyrazolidin-1-ium-2-ide (**2e**). Prepared from **1e** (76%). The $^1$H-NMR data is in agreement with literature. [8]

(*Z*)-5,5-Dimethyl-1-(4-nitrobenzylidene)-3-oxopyrazolidin-1-ium-2-ide (**2f**). Prepared from **1f** (34%). The $^1$H-NMR data is in agreement with literature. [9]

(*Z*)-1-(2,6-Dichlorobenzylidene)-5,5-dimethyl-3-oxopyrazolidin-1-ium-2-ide (**2g**). Prepared from **1g** (88%). The $^1$H-NMR data is in agreement with literature. [9]

(*Z*)-5,5-Dimethyl-3-oxo-1-(3,4,5-trimethoxybenzylidene)pyrazolidin-1-ium-2-ide (**2h**). Prepared from **1h** (71%). The $^1$H-NMR data is in agreement with literature. [9]

(*Z*)-5,5-Dimethyl-1-(naphthalen-2-ylmethylene)-3-oxopyrazolidin-1-ium-2-ide (**2i**). Prepared from **1i** (73%). The $^1$H-NMR data is in agreement with literature. [10]

(*Z*)-1-Benzylidene-4-(((benzyloxy)carbonyl)amino)-5,5-dimethyl-3-oxopyrazolidin-1-ium-2-ide (**2j**). Prepared from **1j** (44%). The $^1$H-NMR data is in agreement with literature. [5]

(*Z*)-5,5-Dimethyl-3-oxo-1-(pyridin-3-ylmethylene)pyrazolidin-1-ium-2-ide (**2k**). Prepared from **1k**. Pale yellow solid (40%); mp 197–199 °C; $\nu_{max}$/cm$^{-1}$ (ATR) 2978, 1665, 1590, 1579, 1559, 1477, 1468, 1431, 1397, 1380, 1295, 1247, 1232, 1143, 1094, 1026, 948, 866, 849, 810, 705, 672, 638, 611; $\delta_H$ (500 MHz; CDCl$_3$; Me$_4$Si) 1.75 (s, 6H), 2.73 (s, 2H), 7.22 (s, 1H), 7.38 (dd, $J$ = 8.2, 4.8 Hz, 1H), 8.59 (dd, $J$ = 4.8, 1.7 Hz, 1H), 8.99 (d, $J$ = 2.2 Hz, 1H), 9.17 (dt, $J$ = 8.4, 2.0 Hz, 1H); $\delta_C$ (126 MHz; CDCl$_3$; Me$_4$Si) 28.9, 44.3, 74.6, 123.7, 126.3, 126.4, 138.0, 151.4, 152.1, 182.0; HRMS (ESI): MH$^+$, found 204.1130. [C$_{11}$H$_{14}$N$_3$O]$^+$ requires 204.1131.

(*Z*)-1-(Furan-2-ylmethylene)-5,5-dimethyl-3-oxopyrazolidin-1-ium-2-ide (**2l**). Prepared from **1l**. Off-white solid (42%); mp 137–145 °C; $\nu_{max}$/cm$^{-1}$ (ATR) 3097, 1664, 1595, 1558, 1479, 1467, 1427, 1408, 1392, 1311, 1272, 1204, 1143, 1097, 1002, 938, 884, 832, 816, 767, 672, 652, 625; $\delta_H$ (500 MHz; CDCl$_3$; Me$_4$Si) 1.67 (s, 6H), 2.73 (s, 2H), 6.63 (ddd, *J* = 3.7, 1.8, 0.7 Hz, 1H), 7.18 (s, 1H), 7.59 (dd, *J* = 1.8, 0.7 Hz, 1H), 7.91 (d, *J* = 3.6 Hz, 1H); $\delta_C$ (126 MHz; CDCl$_3$; Me$_4$Si) 28.8, 44.9, 72.4, 113.7, 118.6, 121.1, 146.2, 146.3, 181.4; HRMS (ESI): MH$^+$, found 193.0976. [C$_{10}$H$_{13}$N$_2$O$_2$]$^+$ requires 193.0972.

(*Z*)-1-Ethylidene-5,5-dimethyl-3-oxopyrazolidin-1-ium-2-ide (**2m**). Prepared from **1m** (47%). The $^1$H-NMR data is in agreement with literature. [11]

**Pyrazolo[1,2-*a*]pyrazoles 3.**

Methyl 1-(4-chlorophenyl)-7,7-dimethyl-5-oxo-6,7-dihydro-1*H*,5*H*-pyrazolo[1,2-*a*]pyrazole-2-carboxylate (**3a**). Prepared from **1c** (80%). The $^1$H-NMR data is in agreement with literature. [12]

Methyl 7,7-dimethyl-5-oxo-1-(3,4,5-trimethoxyphenyl)-6,7-dihydro-1*H*,5*H*-pyrazolo[1,2-*a*]pyrazole-2-carboxylate (**3b**). Prepared from **1h** (82%). The $^1$H-NMR data is in agreement with literature. [12]

Methyl 7,7-dimethyl-1-(4-nitrophenyl)-5-oxo-6,7-dihydro-1*H*,5*H*-pyrazolo[1,2-*a*]pyrazole-2-carboxylate (**3c**). Prepared from **1f** (73%). The $^1$H-NMR data is in agreement with literature. [12]

tert-Butyl ((*S*)-1-((1*S*)-7,7-dimethyl-1-(4-nitrophenyl)-5-oxo-6,7-dihydro-1*H*,5*H*-pyrazolo[1,2-*a*]pyrazol-2-yl)-1-oxopropan2-yl)carbamate (**3d**). Prepared from **1f** (30%). The $^1$H-NMR data is in agreement with literature. [13]

Methyl (1*S*\*,6*R*\*,7*R*\*)-6-benzyloxycarbonylamino-7-isopropyl-1-(4-nitrophenyl)-5-oxo-6,7-dihydro-1*H*,5*H*-pyrazolo[1,2-*a*]pyrazole-2-carboxylate (**3e**). Prepared from **1n** (52%). The $^1$H-NMR data is in agreement with literature. [14]

Methyl 7,7-dimethyl-5-oxo-1-phenethyl-6,7-dihydro-1*H*,5*H*-pyrazolo[1,2-*a*]pyrazole-2-carboxylate (**3f**). Prepared from **1o**. Yellow resin (62%); $\nu_{max}$/cm$^{-1}$ (ATR) 2951, 1732, 1693, 1601, 1496, 1453, 1399, 1331, 1261, 1202, 1117, 1037, 997, 968, 890, 824, 746, 699; $\delta_H$ (500 MHz; CDCl$_3$; Me$_4$Si) 1.05 (s, 3H), 1.33 (s, 3H), 1.93 (dddd, *J* = 14.0, 10.9, 5.8, 3.4 Hz, 3H), 2.06 (ddt, *J* = 13.7, 10.6, 6.0 Hz, 1H), 2.32 (d, *J* = 15.5 Hz, 1H), 2.72 (qdd, *J* = 13.7, 10.5, 5.7Hz, 2H), 2.83 (d, *J* = 15.6 Hz, 1H), 3.73 (s, 3H), 4.54 (ddd, *J* = 6.0, 3.3, 1.3 Hz, 1H), 7.14 – 7.22 (m, 3H), 7.24 – 7.30 (m, 2H), 7.46 (d, *J* = 1.3 Hz, 1H); $\delta_C$ (126 MHz; CDCl$_3$; Me$_4$Si) 18.3, 25.0, 30.9, 37.0, 48.8, 51.5, 60.3, 65.0, 115.2, 125.7, 128.3, 128.4, 131.8, 141.9, 164.4, 168.6; HRMS (ESI): MH$^+$, found 315.1704. [C$_{15}$H$_{17}$N$_2$O$_3$]$^+$ requires 315.1703.

Methyl 1,7,7-trimethyl-5-oxo-6,7-dihydro-1*H*,5*H*-pyrazolo[1,2-*a*]pyrazole-2-carboxylate (**3g**). Prepared from **1m**. Yellow solid (77%); mp 90–93 °C; $\nu_{max}$/cm$^{-1}$ (ATR) 3089, 2977, 1730, 1685, 1590, 1443, 1367, 1327, 1271, 1230, 1190, 1169, 1120, 1095, 1065, 1051, 999, 971, 931, 889, 777, 759, 708; $\delta_H$ (500 MHz; CDCl$_3$; Me$_4$Si) 1.11 (s, 3H), 1.35 (s, 3H), 1.41 (d, *J* = 6.2 Hz, 3H), 2.34 (d, *J* = 15.6 Hz, 1H), 2.82 (d, *J* = 15.6 Hz, 1H), 3.75 (s, 3H), 4.54 (qd, *J* = 6.2, 1.5 Hz, 1H), 7.40 (d, *J* = 1.4 Hz, 1H); $\delta_C$ (126 MHz; CDCl$_3$; Me$_4$Si) 18.8, 23.2, 25.0, 49.3, 51.5, 56.3, 64.4, 117.5, 130.0, 164.6, 166.9; HRMS (ESI): MH$^+$, found 225.1230. [C$_{11}$H$_{17}$N$_2$O$_3$]$^+$ requires 225.1234.

Methyl 1-ethyl-7,7-dimethyl-5-oxo-6,7-dihydro-1*H*,5*H*-pyrazolo[1,2-*a*]pyrazole-2-carboxylate (**3h**). Prepared from **1p**. Yellow resin (51%); $\nu_{max}$/cm$^{-1}$ (ATR) 2965, 1732, 1692, 1601, 1386, 1332, 1310, 1284, 1252, 1200, 1118, 1096, 1023, 963, 920, 891, 826, 754, 705; $\delta_H$ (500 MHz; CDCl$_3$; Me$_4$Si) 0.93 (t, *J* = 7.4 Hz, 3H), 1.06 (s, 3H), 1.32 (s, 3H), 1.59 – 1.77 (m, 2H), 2.31 (d, *J* = 15.4 Hz, 1H), 2.81 (d, *J* = 15.5 Hz, 1H), 3.74 (s, 3H), 4.42 (ddd, *J* = 6.0, 3.4, 1.3 Hz, 1H), 7.45 (d, *J* = 1.3 Hz, 1H); $\delta_C$ (126 MHz; CDCl$_3$; Me$_4$Si) 9.0, 18.2,

24.9, 28.2, 48.8, 51.5, 61.5, 65.0, 115.2, 131.8, 164.6, 168.8; HRMS (ESI): MH⁺, found 239.1388. [$C_{12}H_{19}N_2O_3$]⁺ requires 239.1390.

Methyl (1*S*\*,7*R*\*)-1-methyl-5-oxo-7-phenyl-6,7-dihydro-1*H*,5*H*-pyrazolo[1,2-*a*]pyrazole-2-carboxylate (**3i**). Prepared from **1q**. Orange resin (34%); $\nu_{max}$/cm⁻¹ (ATR) 1691, 1598, 1449, 1403, 1371, 1336, 1300, 1263, 1208, 1164, 1099, 1062, 953, 920, 836, 757, 700, 676, 637; $\delta_H$ (500 MHz; CDCl₃; Me₄Si) 1.18 (d, *J* = 6.3 Hz, 3H), 2.90 (dd, *J* = 16.4, 6.6 Hz, 1H), 3.00 (dd, *J* = 16.4, 12.6 Hz, 1H), 3.75 (s, 3H), 4.25 (dd, *J* = 12.6, 6.7 Hz, 1H), 4.30 (qd, *J* = 6.3, 1.7 Hz, 1H), 7.35 – 7.43 (m, 3H), 7.45 (d, *J* = 1.7 Hz, 1H), 7.47 – 7.51 (m, 2H); $\delta_C$ (126 MHz; CDCl₃; Me₄Si) 20.9, 44.6, 51.5, 65.5, 71.4, 118.2, 127.7, 128.9, 129.3, 136.7, 164.3, 165.5; HRMS (ESI): MH⁺, found 273.1227. [$C_{15}H_{17}N_2O_3$]⁺ requires 273.1234.

Methyl (1*S*\*,7*R*\*)-1,7-dimethyl-5-oxo-6,7-dihydro-1*H*,5*H*-pyrazolo[1,2-*a*]pyrazole-2-carboxylate (**3j**). Prepared from **1r**. Yellow resin (25%); $\nu_{max}$/cm⁻¹ (ATR) 2974, 1693, 1599, 1446, 1405, 1381, 1335, 1308, 1221, 1183, 1152, 1103, 1076, 1049, 982, 929, 872, 785, 758, 719, 673; $\delta_H$ (500 MHz; CDCl₃; Me₄Si) 1.33 (d, *J* = 6.1 Hz, 3H), 1.49 (d, *J* = 6.3 Hz, 3H), 2.56 – 2.68 (m, 2H), 3.24 – 3.35 (m, 1H), 3.75 (s, 3H), 4.26 (qd, *J* = 6.3, 1.7 Hz, 1H), 7.36 (d, *J* = 1.6 Hz, 1H); $\delta_C$ (126 MHz; CDCl₃; Me₄Si) 17.5, 22.0, 42.8, 51.5, 63.4, 65.0, 117.8, 129.5, 164.4, 166.6; HRMS (ESI): MH⁺, found 211.1078. [$C_{10}H_{15}N_2O_3$]⁺ requires 211.1077.

(3*R*\*,5*S*\*)-6-Acetyl-3,5-dimethyl-2,3-dihydro-1*H*,5*H*-pyrazolo[1,2-*a*]pyrazol-1-one (**3k**). Prepared from **1r** and 3-butyn-2-one. Yellow solid (16%); mp 128–130 °C; $\nu_{max}$/cm⁻¹ (ATR) 2964, 2925, 1722, 1643, 1540, 1413, 1259, 1192, 1091, 1018, 797; $\delta_H$ (500 MHz; CDCl₃; Me₄Si) 1.26 (d, *J* = 6.1 Hz, 3H), 1.39 (d, *J* = 6.3 Hz, 3H), 2.21 (s, 3H), 2.50 – 2.62 (m, 2H), 3.15 – 3.25 (m, 1H), 4.22 (qd, *J* = 6.4, 1.5 Hz, 1H), 7.28 (d, *J* = 1.3 Hz, 1H); $\delta_C$ (126 MHz; CDCl₃; Me₄Si) 17.5, 21.9, 26.8, 42.7, 63.5, 65.1, 127.3, 129.7, 167.6, 193.5; HRMS (ESI): MH⁺, found 195.1127. [$C_{10}H_{15}N_2O_2$]⁺ requires 195.1128.

(3*R*\*,5*S*\*)-6-Benzoyl-3,5-dimethyl-2,3-dihydro-1*H*,5*H*-pyrazolo[1,2-*a*]pyrazol-1-one (**3l**). Prepared from **1r** and 1-phenyl-2-propyn-1-one. Yellow resin (11%); $\nu_{max}$/cm⁻¹ (ATR) 2973, 1715, 1622, 1568, 1408, 1382, 1357, 1308, 1268, 1217, 1177, 1135, 1078, 1016, 979, 943, 867, 795, 750, 713, 661, 613; $\delta_H$ (500 MHz; CDCl₃; Me₄Si) 1.38 (d, *J* = 6.1 Hz, 3H), 1.56 (d, *J* = 6.3 Hz, 3H), 2.60 – 2.73 (m, 2H), 3.33 (ddq, *J* = 12.0, 7.2, 6.1 Hz, 1H), 4.54 (qd, *J* = 6.3, 1.5 Hz, 1H), 7.20 (d, *J* = 1.4 Hz, 1H), 7.46 (dd, *J* = 8.3, 7.0 Hz, 2H), 7.53 – 7.58 (m, 1H), 7.69 – 7.74 (m, 2H); $\delta_C$ (126 MHz; CDCl₃; Me₄Si) 17.6, 21.4, 42.8, 63.5, 66.5, 126.4, 128.3, 128.6, 131.0, 132.3, 138.7, 167.0, 191.0; HRMS (ESI): MH⁺, found 257.1290. [$C_{15}H_{17}N_2O_2$]⁺ requires 257.1285.

**Pyrazolidin-3-ones 4 and pyrazolones 5.**

Compounds **4a-e** were prepared according to established literature procedures [15-17].

1-Phenylpyrazolidin-3-one (**4a**). The ¹H-NMR data is in agreement with literature. [15]

1-(4-Chlorophenyl)pyrazolidin-3-one (**4b**). The ¹H-NMR data is in agreement with literature. [15]

1-(4-Methoxyphenyl)pyrazolidin-3-one (**4c**). The ¹H-NMR data is in agreement with literature. [16]

5-Methyl-1-phenylpyrazolidin-3-one (**4d**). The ¹H-NMR data is in agreement with literature. [15]

1,5-Diphenylpyrazolidin-3-one (**4e**). The ¹H-NMR data is in agreement with literature. [17]

1-Phenyl-2*H*-pyrazolin-3-one (**5a**). Prepared from **1s** (58%). The ¹H-NMR data is in agreement with literature. [18]

1-(4-Chlorophenyl)-2*H*-pyrazolin-3-one (**5b**). Prepared from **1t** (63%). The ¹H-NMR data is in agreement with literature. [19]

1-(4-Methoxyphenyl)-2*H*-pyrazolin-3-one (**5c**). Prepared from **1u** (84%). The characterization data is in agreement with literature. [20]

5-Methyl-1-phenyl-2*H*-pyrazolin-3-one (**5d**). Prepared from **1v** (30%). The [1]H-NMR data is in agreement with literature. [21]

1,5-Diphenyl-2*H*-pyrazolin-3-one (**5e**). Prepared from **1w** (20%). The [1]H-NMR data is in agreement with literature. [22]

5. ¹H and ¹³C NMR spectra of novel compounds **1**, **2** and **3**.

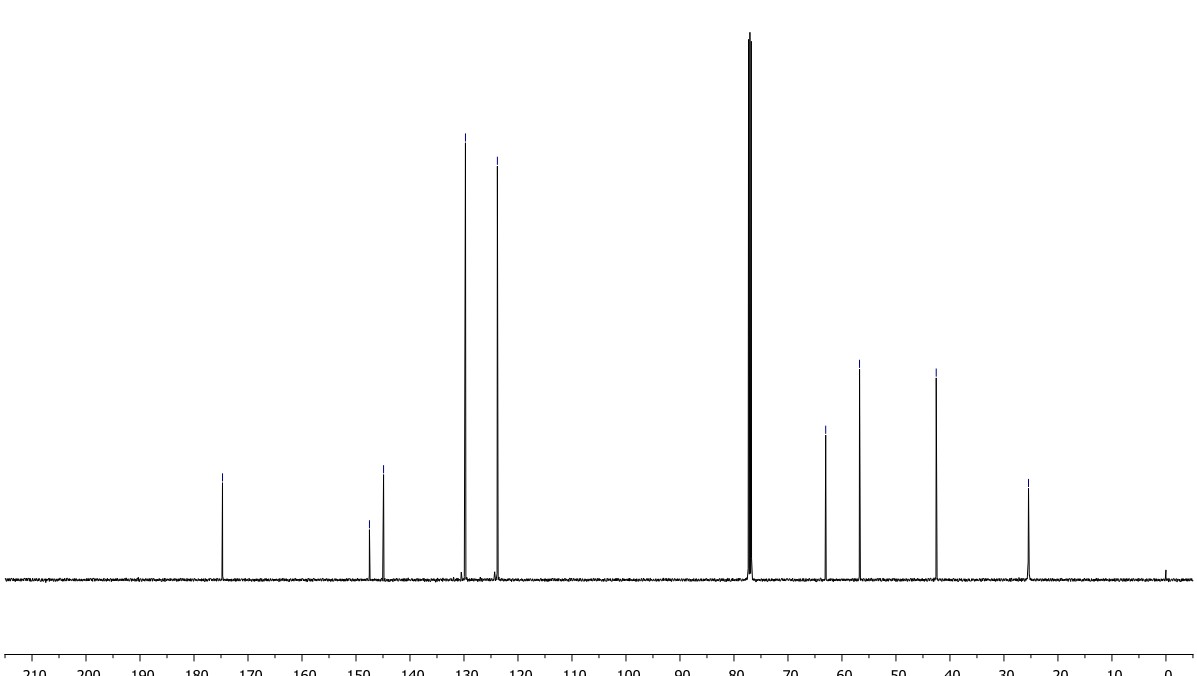

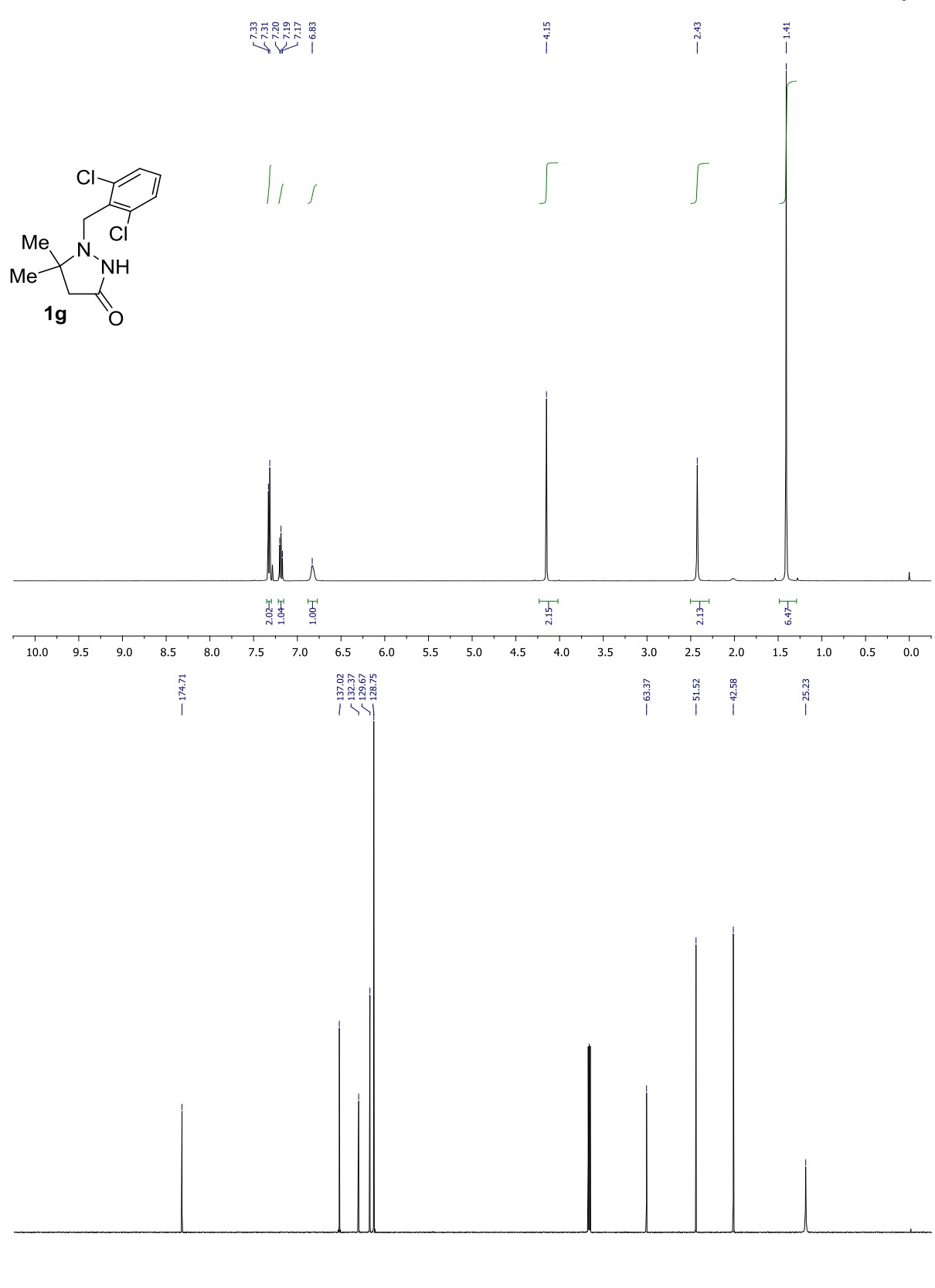

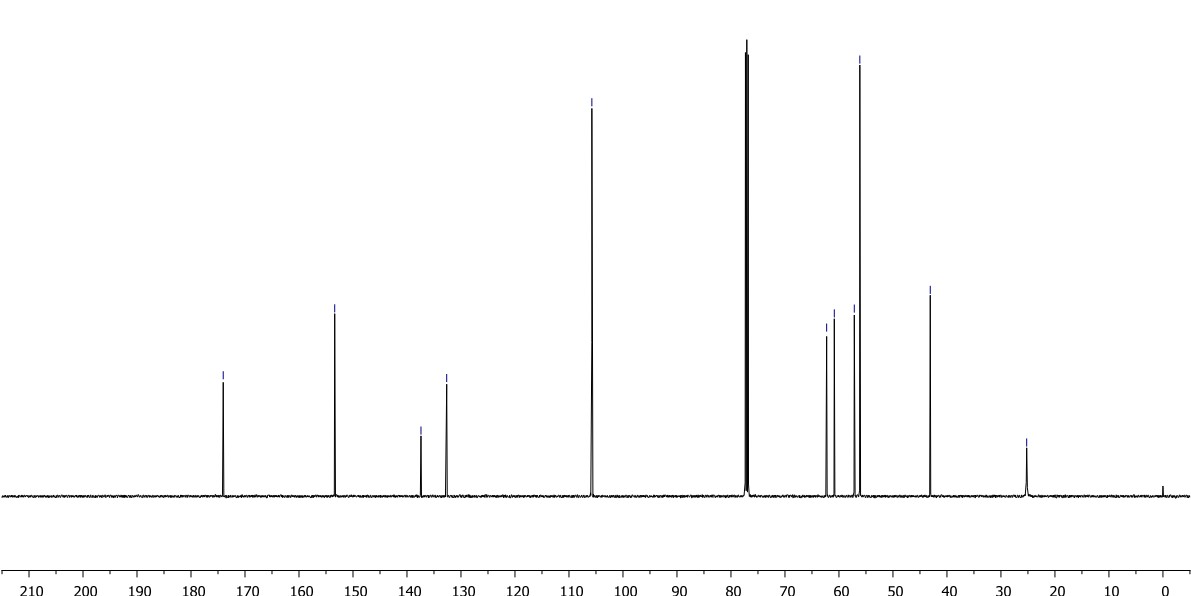

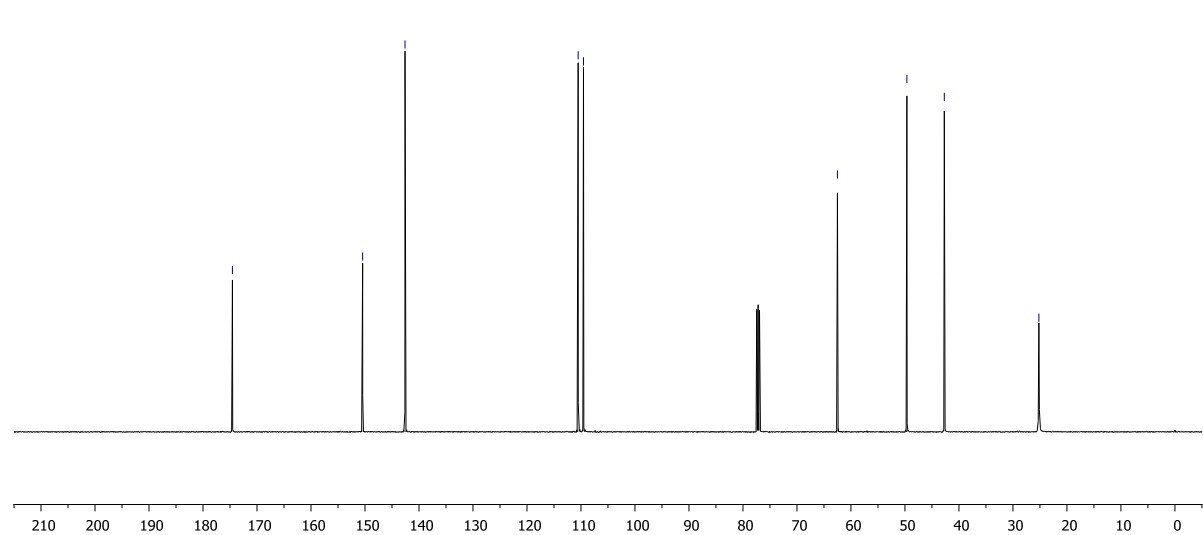

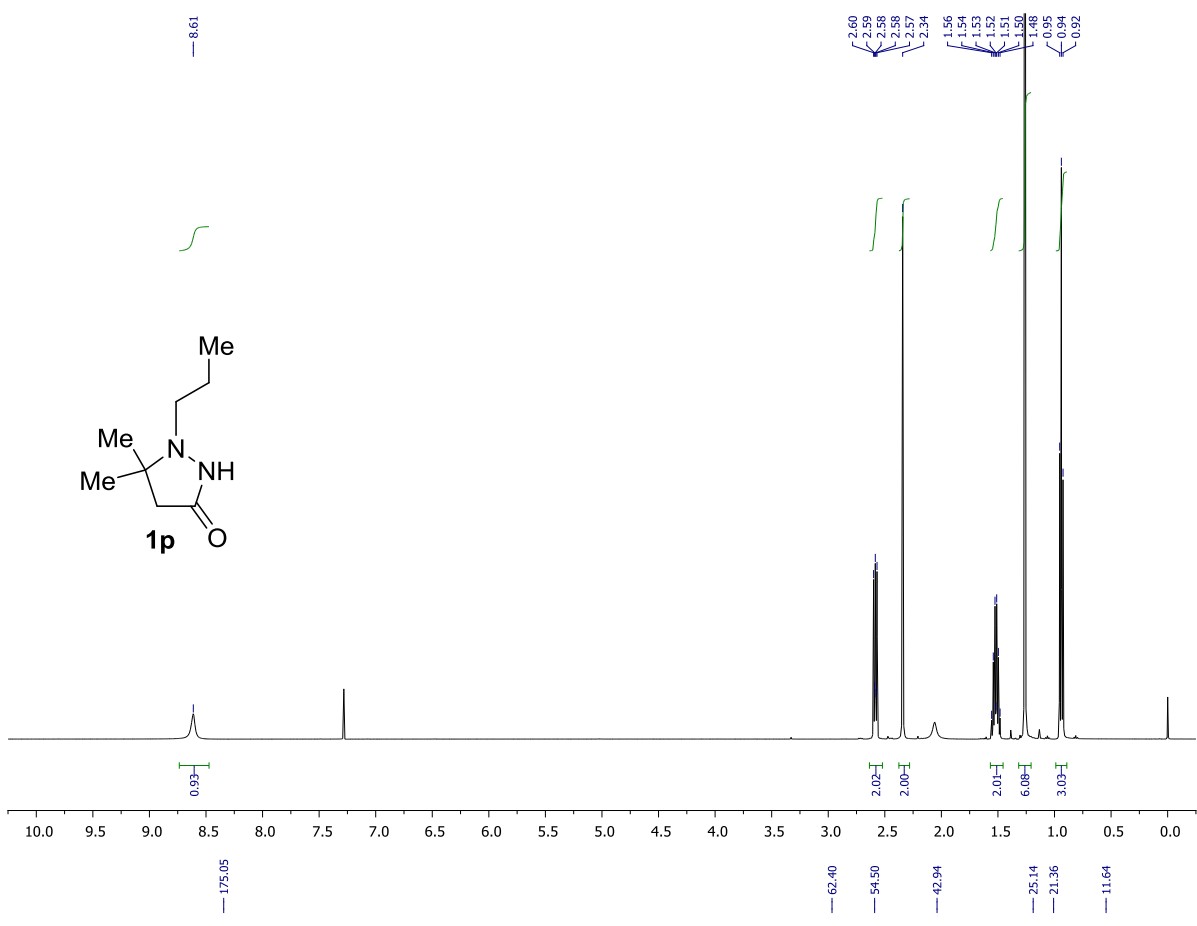

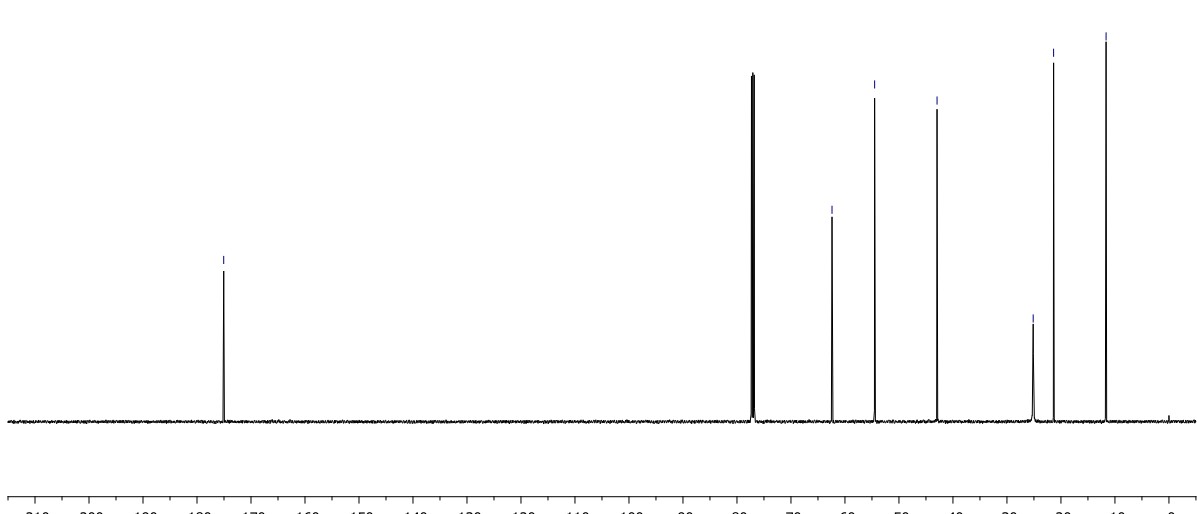

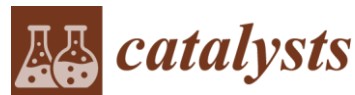

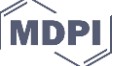

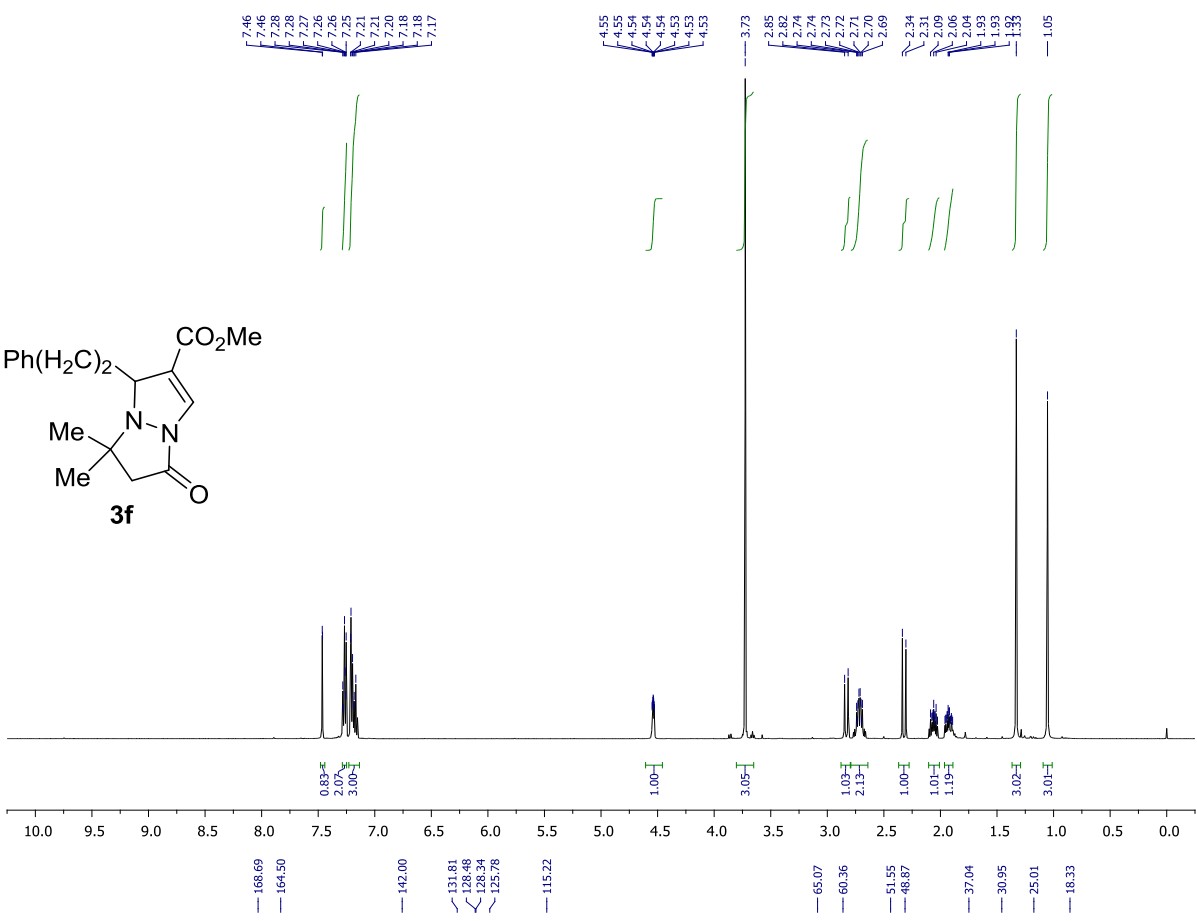

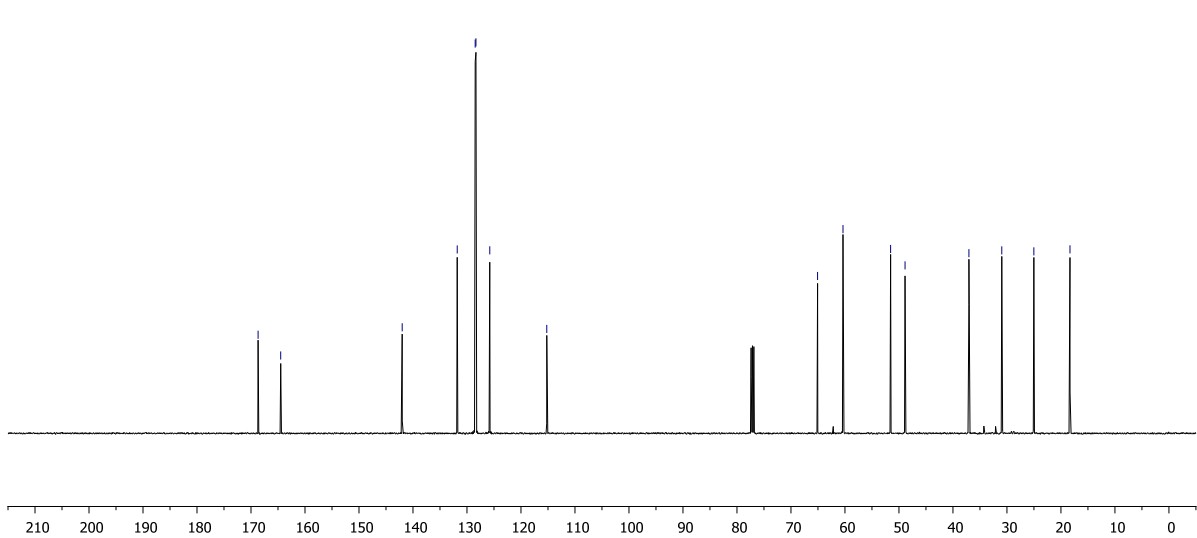

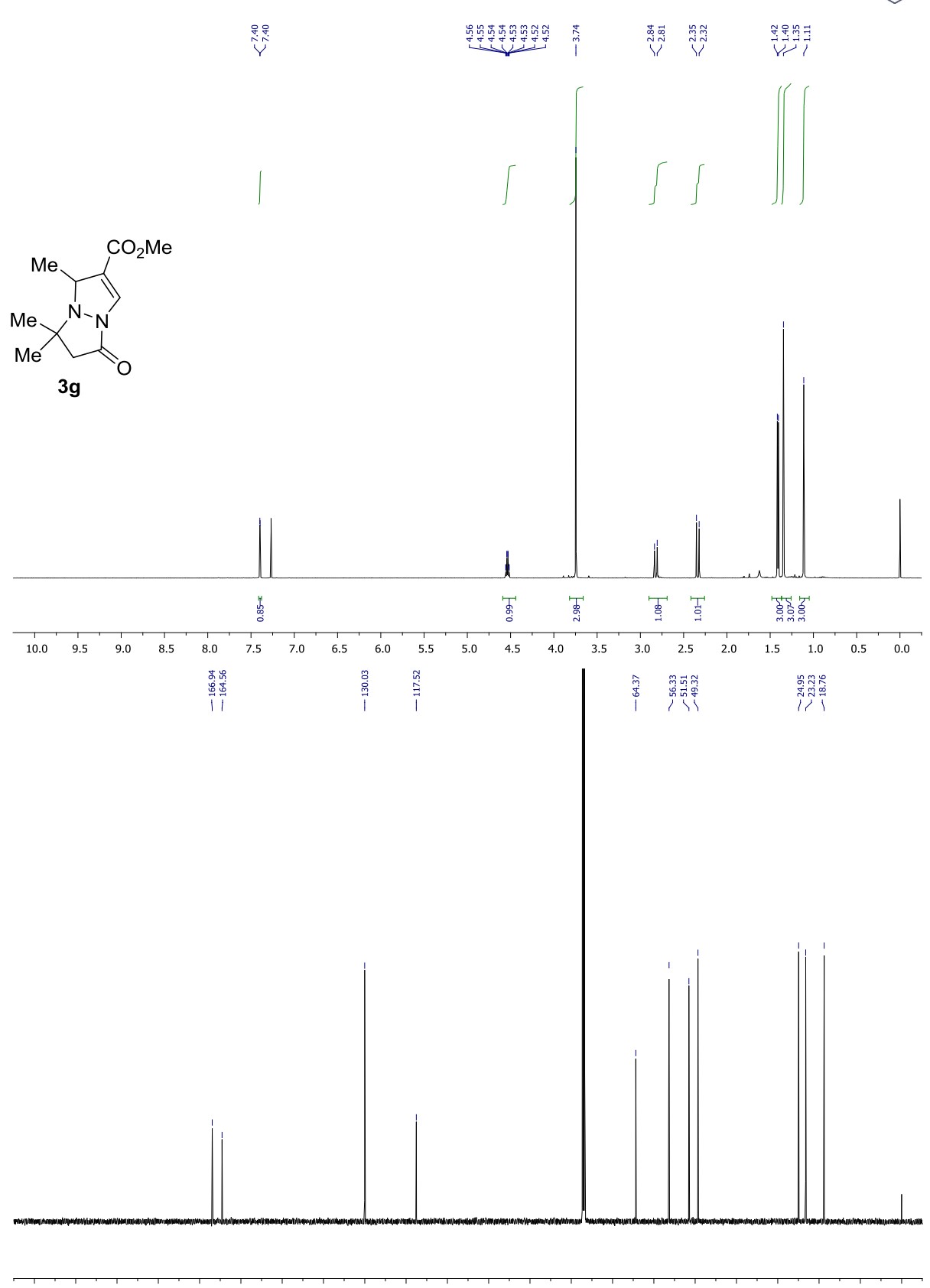

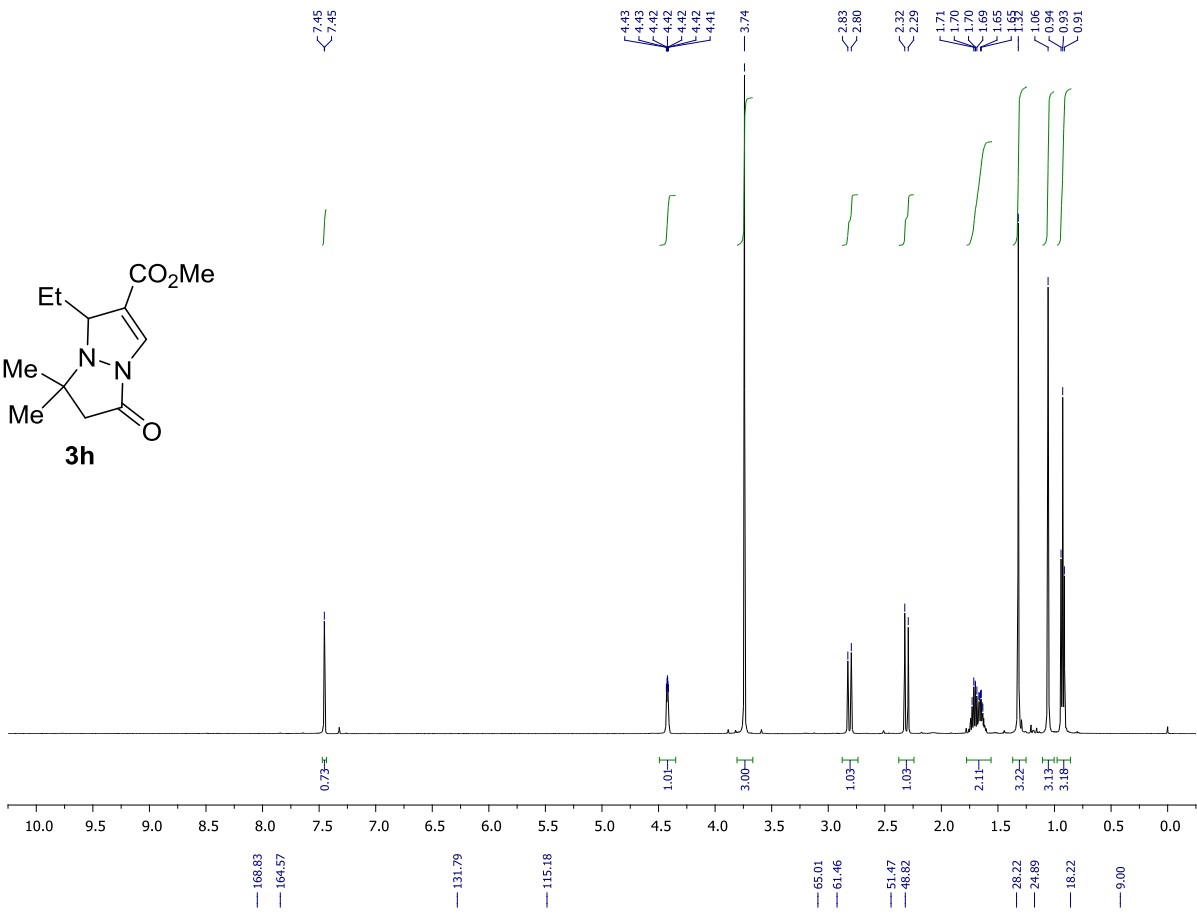

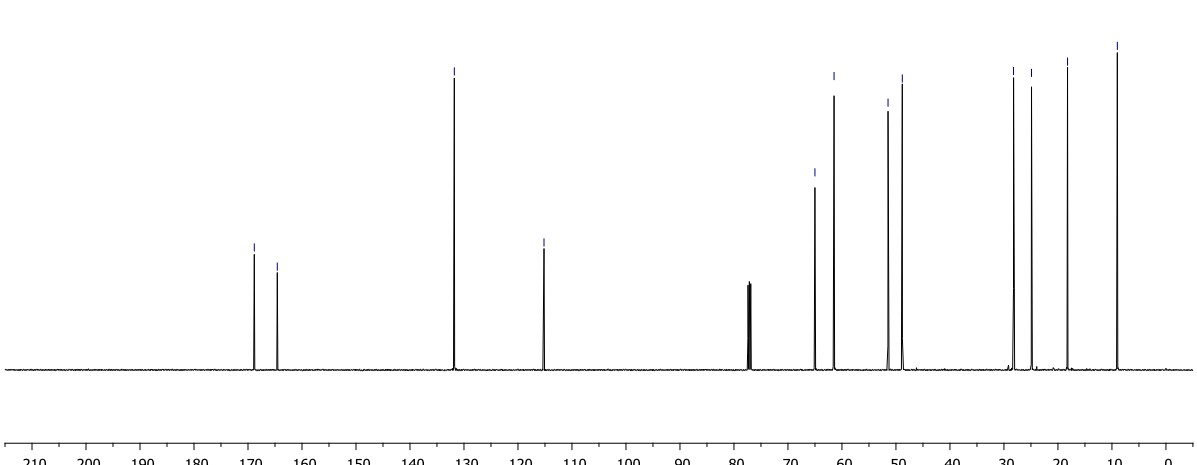

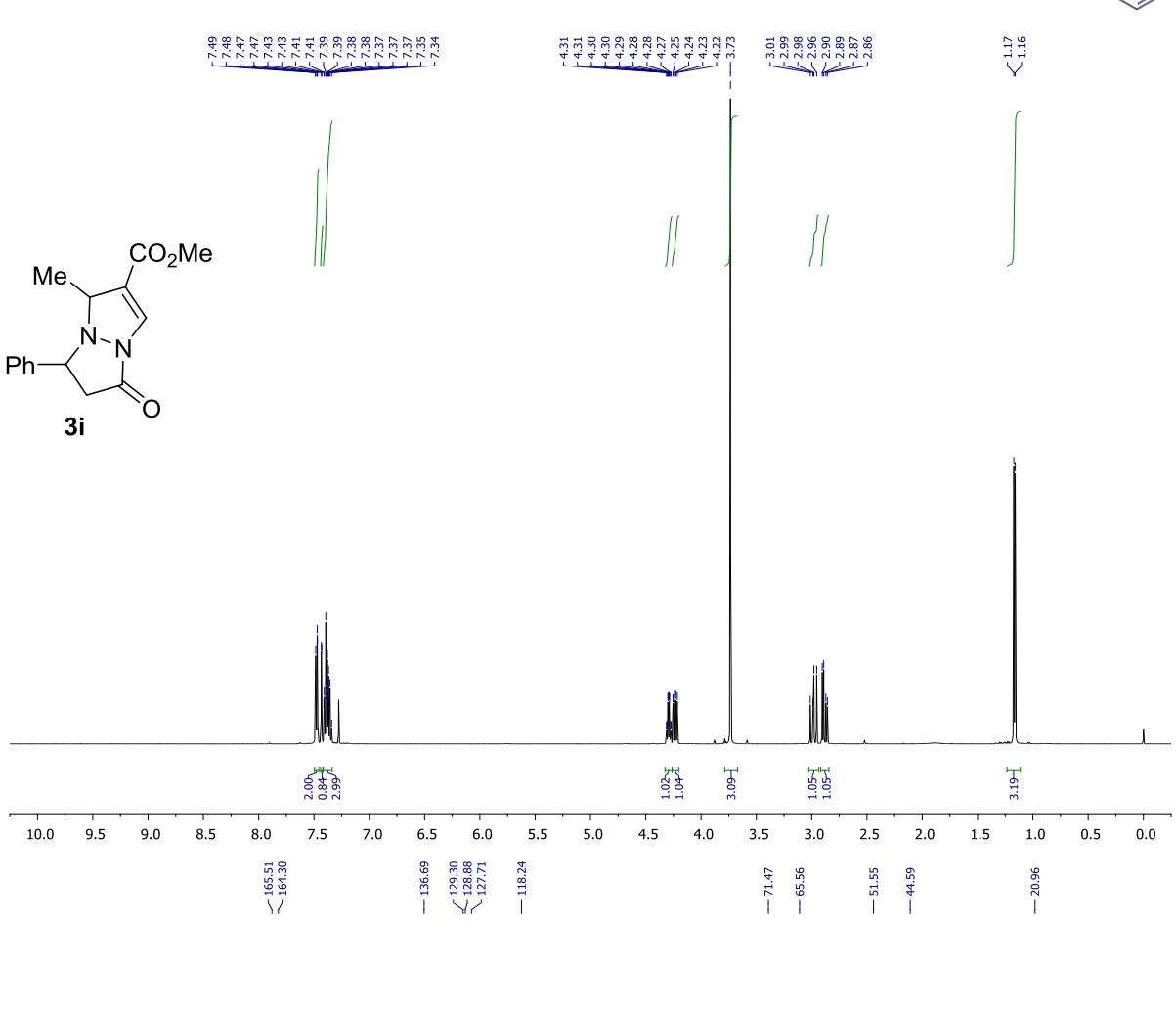

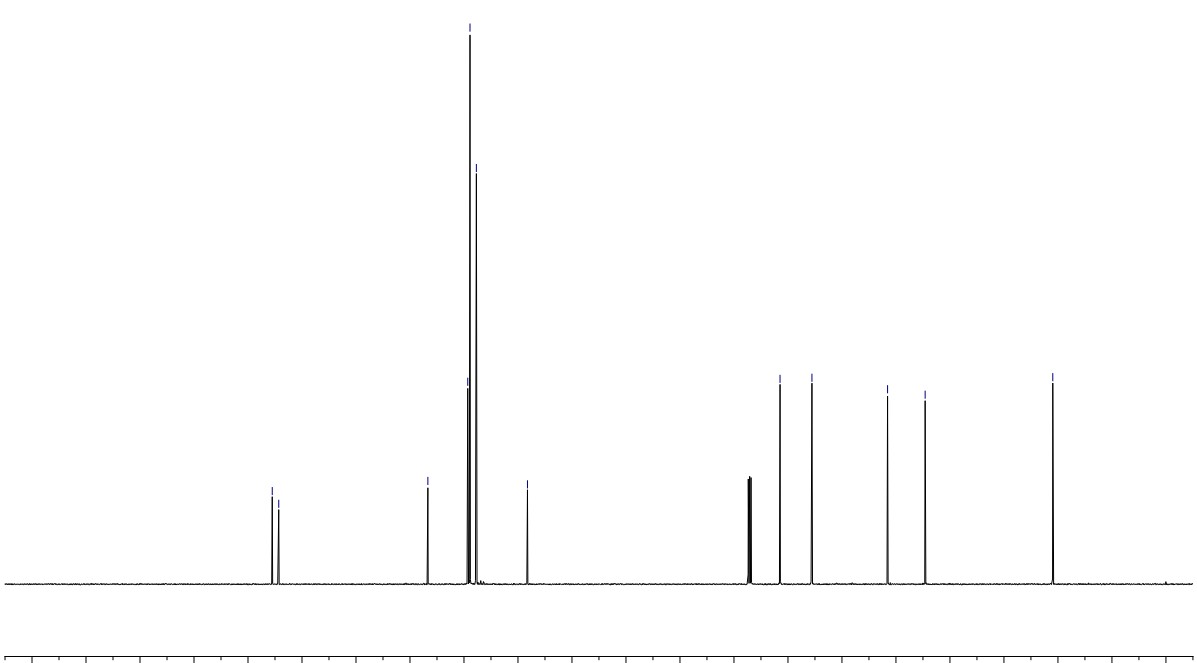

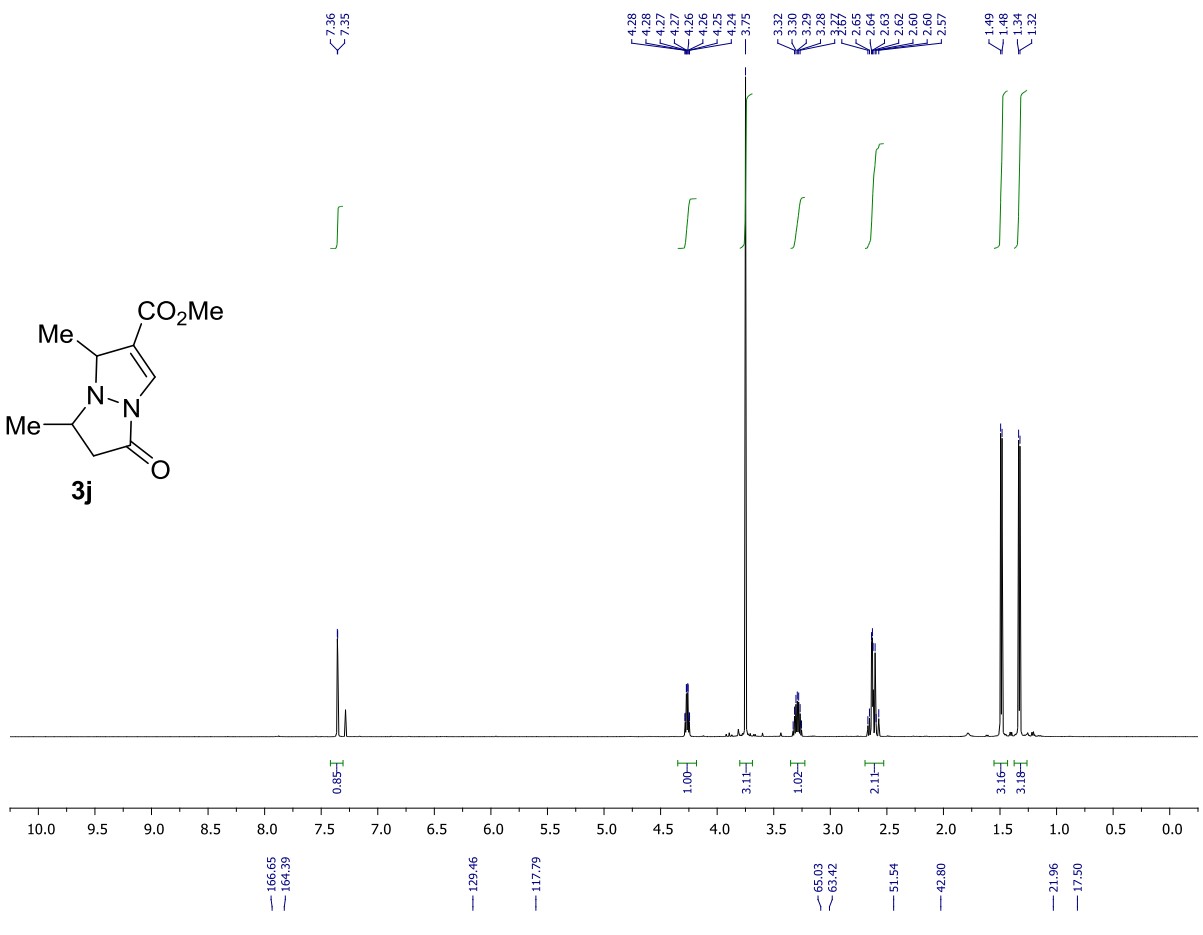

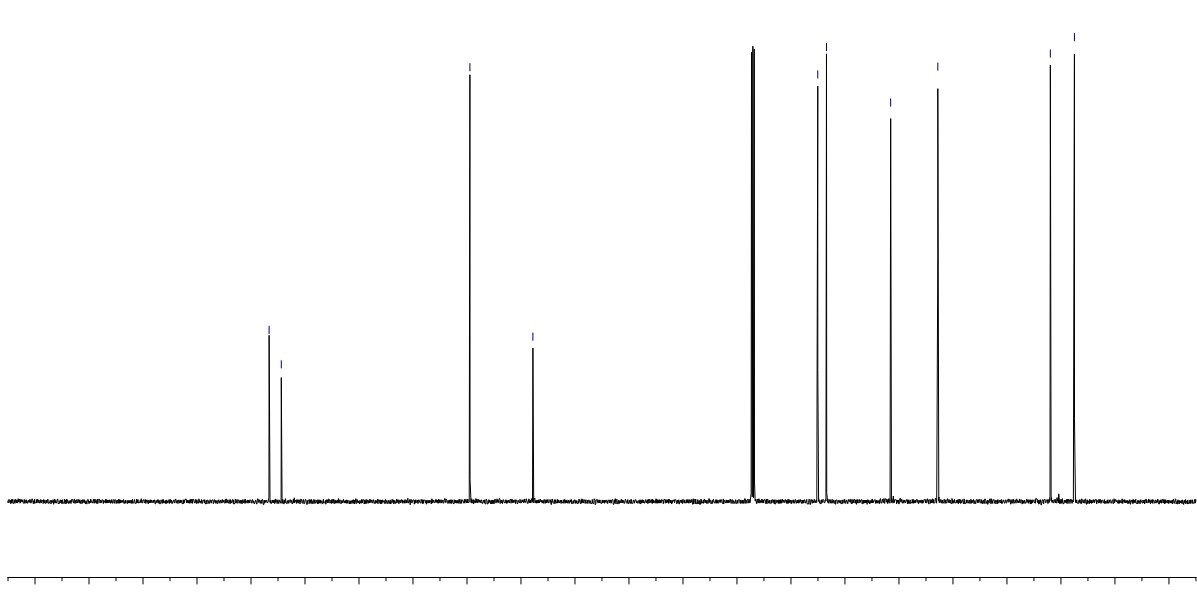

7.28
7.28

4.24
4.23
4.22
4.21
4.20
4.20

3.22
3.21
3.20
3.20
3.19
3.18

2.57
2.57
2.56
2.55
2.21

1.39
1.38
1.26
1.25

COMe

Me

Me

**3k**

O

N N

0.95

1.00

1.02

2.09

3.15

3.20
3.30

10.0  9.5  9.0  8.5  8.0  7.5  7.0  6.5  6.0  5.5  5.0  4.5  4.0  3.5  3.0  2.5  2.0  1.5  1.0  0.5  0.0

193.54

167.58

129.69
127.34

65.05
63.50

42.73

26.82
21.93
17.45

210  200  190  180  170  160  150  140  130  120  110  100  90  80  70  60  50  40  30  20  10  0

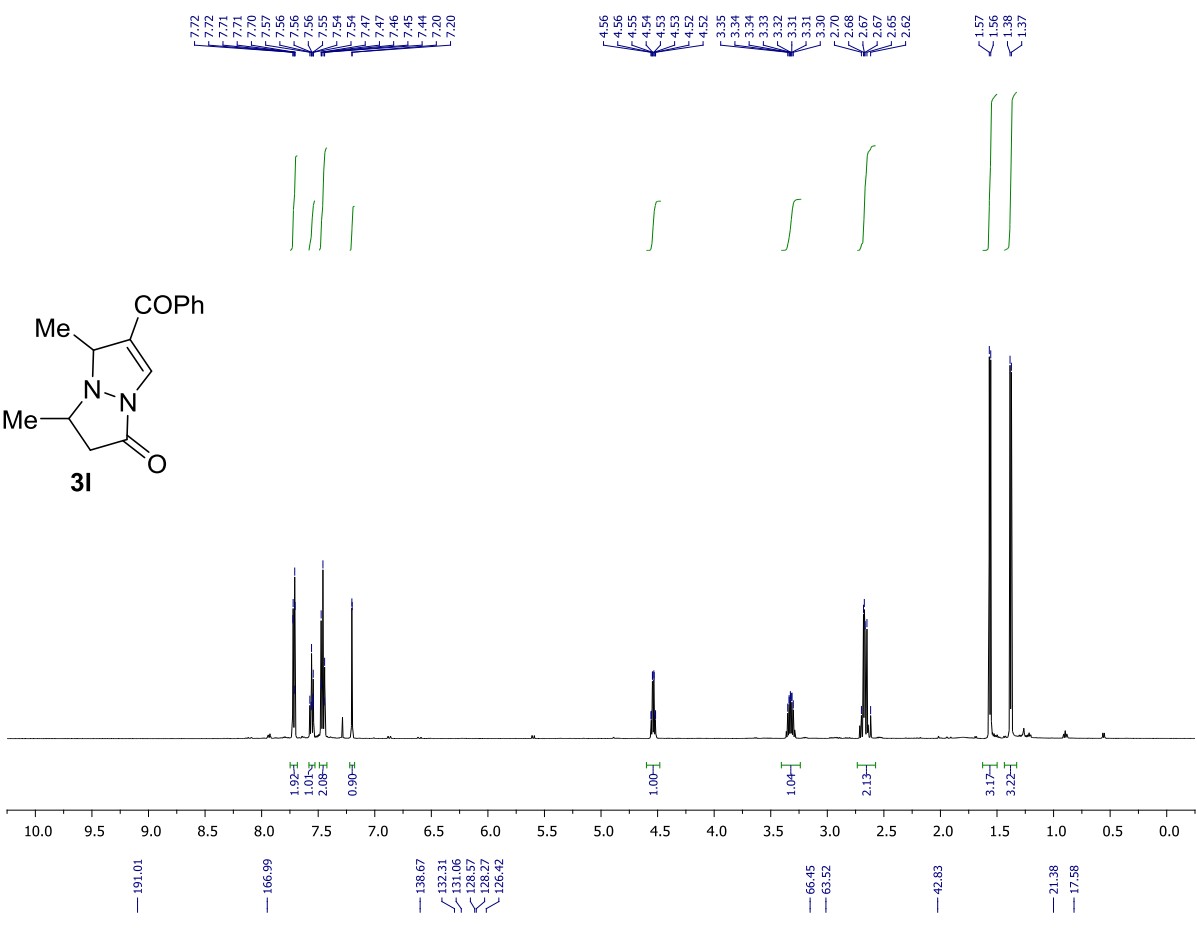

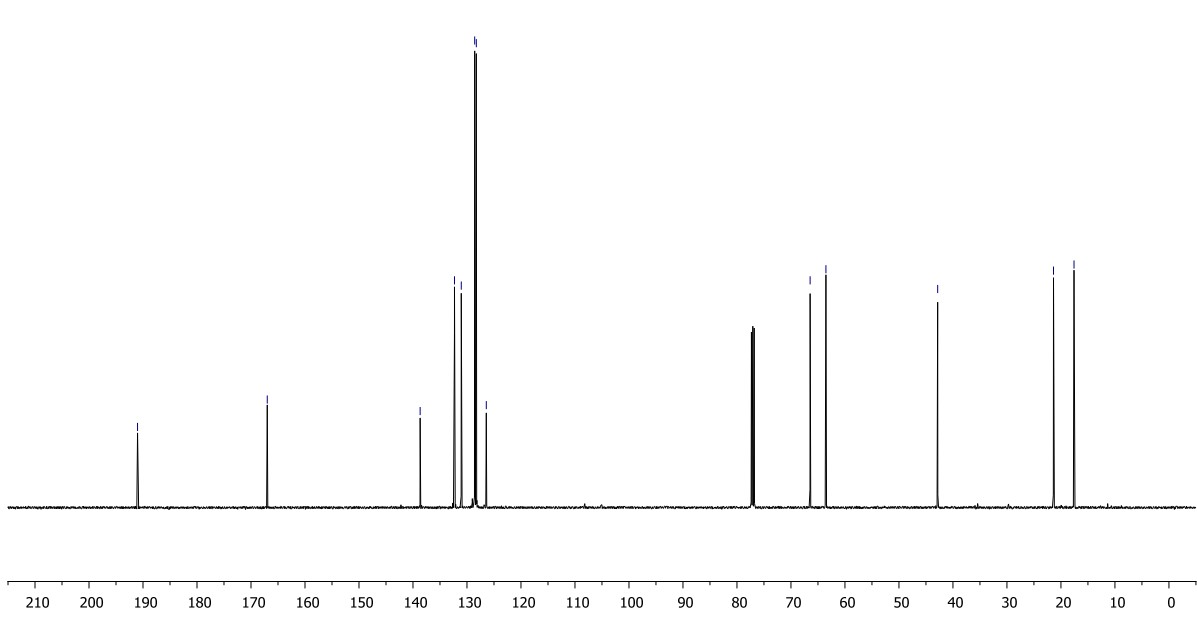

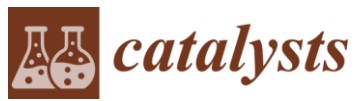

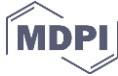

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
