# Peer review of "Eosin Y-Catalyzed Visible-Light-Mediated Aerobic Transformation of Pyrazolidine-3-One Derivatives"

_catalysts, doi:10.3390/catal10090981_

Round 1

Reviewer 1 Report

This is an interesting method for pyrazolopyrazoles relying on photooxidation to generate azomethine imines that then are either isolated or undergo 1,3-dipolar cycloaddition to acetylenes. the yields are good to fair, perhaps the authors should pay more attention to compare the current method to existing methods like the CuAIAC they have themselves reported on. otherwise I think this is a nice addition to the existing literature and it can be more or less accepted as it is but it should be checked for language and typo's removed like pirazol- should everywhere be pyrazol- etc...

Author Response

Reviewer 1

This is an interesting method for pyrazolopyrazoles relying on photooxidation to generate azomethine imines that then are either isolated or undergo 1,3-dipolar cycloaddition to acetylenes. the yields are good to fair, perhaps the authors should pay more attention to compare the current method to existing methods like the CuAIAC they have themselves reported on. otherwise I think this is a nice addition to the existing literature and it can be more or less accepted as it is but it should be checked for language and typo's removed like pirazol- should everywhere be pyrazol- etc...

Comments

1) -but it should be checked for language and typo's removed like pirazol- should everywhere be pyrazol- etc...

Response

The manuscript was checked and proofread by English native. The spelling and grammar errors were corrected and are marked in yellow in main text.

Reviewer 2 Report

This work reports visible light-induced oxidation of pyrazolidine-3-oness and sequential cycloaddition with electron-deficient alkenes using Eosin-Y as an organic photoredox catalyst. Although a generality of substrate is limited, it is very interesting results and the experiments appear to have been very carefully performed. Thus, this might be acceptable for Catalysts. Nevertheless, the author should make a few modifications/additions, as indicated at the end of this text.

1) p2, Scheme 1, In “This Work”, please add “-2e-, -2H+” and delete “[O]” below arrow.

2) p2, line 74, I don’t agree your explanation about necessity of TFA. In this photoreaction, two protons were released for the formation of 2. Why do you need acid? I believe it is related to acidic form of Eosin-Y. You should add explanation or additional experiments about this in the main text.

3) p6, line 177, You should mention the oxidation potential of 1 and reduction potential of the excited triplet state of Eosin-Y for the first electron transfer in the proposed mechanism. If it is difficult to measure CV (cyclic voltammetry) of 1, please research CV of the similar compounds in literature.

Author Response

Reviewer 2

This work reports visible light-induced oxidation of pyrazolidine-3-oness and sequential cycloaddition with electron-deficient alkenes using Eosin-Y as an organic photoredox catalyst. Although a generality of substrate is limited, it is very interesting results and the experiments appear to have been very carefully performed. Thus, this might be acceptable for Catalysts. Nevertheless, the author should make a few modifications/additions, as indicated at the end of this text.

Comments

1) p2, Scheme 1, In “This Work”, please add “-2e-, -2H+” and delete “[O]” below arrow.

Response

The scheme was corrected as suggested by reviewer.

2) p2, line 74, I don’t agree your explanation about necessity of TFA. In this photoreaction, two protons were released for the formation of 2. Why do you need acid? I believe it is related to acidic form of Eosin-Y. You should add explanation or additional experiments about this in the main text.

Response

The additional explanation of TFA function was added according to our experimental findings. Added text is marked in yellow.

3) p6, line 177, You should mention the oxidation potential of 1 and reduction potential of the excited triplet state of Eosin-Y for the first electron transfer in the proposed mechanism. If it is difficult to measure CV (cyclic voltammetry) of 1, please research CV of the similar compounds in literature.

Response

The comments on the oxidation potential of 1 and reduction potential of the excited triplet state of Eosin-Y have been added and the text (line 188-190) is marked in yellow. The CV data of 1 are in the supporting information. The corresponding reference [28] was added as well.